# A Within-Sample Comparison of Two Innovative Neuropsychological Tests for Assessing ADHD

**DOI:** 10.3390/brainsci11010036

**Published:** 2020-12-31

**Authors:** Anna Baader, Behnaz Kiani, Nathalie Brunkhorst-Kanaan, Sarah Kittel-Schneider, Andreas Reif, Oliver Grimm

**Affiliations:** 1Department of Psychiatry, Psychosomatic Medicine and Psychotherapy, University Hospital, Goethe University Frankfurt am Main, 60528 Frankfurt, Germany; anna.baader@kgu.de (A.B.); behnazkiani1990@gmail.com (B.K.); nathalie.brunkhorst-kanaan@kgu.de (N.B.-K.); andreas.reif@kgu.de (A.R.); 2Department of Clinical Psychology, School of Education and Psychology, Shiraz University, Shiraz 71946-84334, Iran; 3Department of Psychiatry, Psychotherapy and Psychosomatic Medicine, University Hospital, University of Würzburg, 97080 Würzburg, Germany; kittel_s@ukw.de

**Keywords:** ADHD, neuropsychology, continuous performance test, Qb-Test, Nesplora Aquarium, attention, hyperactivity, GHQ-28, UPPS, impulsivity

## Abstract

New innovative neuropsychological tests in attention deficit hyperactivity disorder ADHD have been proposed as objective measures for diagnosis and therapy. The current study aims to investigate two different commercial continuous performance tests (CPT) in a head-to-head comparison regarding their comparability and their link with clinical parameters. The CPTs were evaluated in a clinical sample of 29 adult patients presenting in an ADHD outpatient clinic. Correlational analyses were performed between neuropsychological data, clinical rating scales, and a personality-based measure. Though inattention was found to positively correlate between the two tests (*r* = 0.49, *p* = 0.01), no association with clinical measures and inattention was found for both tests. While hyperactivity did not correlate between both tests, current ADHD symptoms were positively associated with Nesplora Aquarium’s motor activity (*r* = 0.52 to 0.61, *p* < 0.05) and the Qb-Test’s hyperactivity (*r* = 0.52 to 0.71, *p* < 0.05). Conclusively, the overall comparability of the tests was limited and correlation with clinical parameters was low. While our study shows some interesting correlation between clinical symptoms and sub-scales of these tests, usage in clinical practice is not recommended.

## 1. Introduction

Attention deficit hyperactivity disorder (ADHD) is one of the most common neurodevelopmental disorders in childhood [1] and persists into adulthood in 40–55% of the cases [2]. The prevalence of ADHD in adults has been estimated to be 3.4% of the general population [3]. In order to diagnose ADHD in adults, a threshold of five inattention and/or five hyperactivity–impulsivity symptoms that cause clinically significant impairment in at least two functional domains of life is required in DSM-5 [4]. Adult ADHD has been associated with comorbid mental disorders such as depression, anxiety disorders, bipolar disorder, and substance use disorders [5,6,7] as well as non-psychiatric problems, for example obesity and traffic accidents [8,9]. Individuals with ADHD experience more functional impairment in multiple domains of life such as education, work, and risky activities [10,11]. To avoid such negative consequences, adults with ADHD require a precise diagnosis to allow specific treatment. According to the current clinical guidelines, continuous performance tests, i.e., neuropsychological tests measuring selective and sustained attention, have no role in the diagnostic process. Diagnosis of ADHD, or the exclusion thereof, should not be made solely based on neuropsychological testing. Neuropsychological testing and behavioral observation can serve a supplementary tool in the diagnostic process [12]. In the British guidelines, neuropsychological testing is not specifically recommended for diagnosis. Rating scales and observational data should be taken into account when there is doubt about the diagnosis [13]. ADHD is a clinical diagnosis made on the basis of the developmental background, medical history as well as former and current symptoms of patients.

Continuous performance tests (CPT) are among neuropsychological tests measuring the core symptoms of ADHD including inattention and impulsivity. Omission errors, commission errors, and hit reaction time are considered to be associated with ADHD symptoms [14,15,16]. However, the accuracy and usefulness of CPTs in ADHD patients are critically discussed due to their lack of sensitivity and specificity [17]. The predictive power of these measures in differentiating ADHD patients from controls, or patients with other mental disorders, is insufficient [18,19]. CPT variables and ADHD symptoms are associated only to a small amount, as only 52% of ADHD patients have correctly been classified as ADHD by using a CPT [20]. With an error rate of 80% in classifying the patients into the inattentive subtype and 23% in classifying the patients into the hyperactive subtype, the CPT was not able to correctly discriminate between the different subtypes of ADHD.

Recent findings suggest that a multidimensional approach using neuropsychological assessments in combination with other clinical measures enhances establishing an adult ADHD diagnosis [21]. An extension of conventional symptom-based classification by integrating different methods and perspectives may enable precise diagnosis, which is needed to initiate accurate treatment [22]. ADHD is a good test field for this endeavor, as neuropsychological tests offer the possibility to overcome clinical problems. Neuropsychological data provides evidence that executive functioning such as response inhibition, vigilance, working memory, and planning is impaired among patients with ADHD [23]. Nevertheless, moderate effect sizes and lack of universality speak against the hypothesis that ADHD solely arises from a deficit in executive control [23]. A role for tests of inhibitory deficits and delay aversion for a more precise definition of the clinical phenotype is proposed [24]. 

Recent commercial neuropsychological tests incorporated neuropsychological testing with an innovative twist. In this study we used two CPT measures including Nesplora Aquarium (Nesplora Technology and Behavior) [25] and the Quantified Behavioral Test (Qb-Test, marketed by Qbtech) [26]. The Qb-Test measures not only inattention and impulsivity but also hyperactivity with the use of an infrared camera and a reflector attached to a headband to record movements during the test. In Nesplora Aquarium [25] a virtual reality (VR) CPT, motor activity is measured through VR optical devices equipped with sensors and headphones. To improve the ecological validity of CPTs, VR CPTs have particularly been designed by providing a better representation of real-life situations [25]. Both tests are marketed by the manufacturer especially for clinicians but are currently not recommended for standard clinical routine according to guidelines. Therefore, the development of valid neuropsychological tools to assess and evaluate symptoms before and after treatment is highly important.

Several studies have compared VR measures with different commercial CPTs (Test of Variables of Attention: TOVA, Conner’s CPT, Vigil CPT). Variables based on VR better differentiate between ADHD subtypes (inattentive, hyperactive/impulsive, combined) and controls than CPTs not involving VR [27]. However, only 57% of ADHD patients and controls were correctly discriminated. In a recent meta-analysis, VR CPTs have been found to better differentiate between ADHD patients and controls based on the evaluation of omission errors and hit reaction time. In comparison, commission errors showed a higher discriminant power in traditional CPTs compared to VR CPTs [16]. In another comparative study, omission errors and the number of correct responses were found to be associated with current and retrospective ADHD symptoms [14]. As the results were found in healthy students, the transferability to ADHD patients is limited and should be reviewed in a clinical sample. 

To our knowledge, a head-to-head comparison of these innovative tests within one same sample has not yet been made but might have important implications for test selection and clinical routing. Therefore, in this study we evaluate Nesplora Aquarium and the Qb-Test in the diagnostic workflow with patients of our adult ADHD outpatient clinic. The first aim of the current study is to compare Nesplora Aquarium and the Qb-Test in a within-sample comparison testing adult patients presenting to an ADHD outpatient clinic. Second, we aim to assess the ecological validity of the two CPTs (i.e., Nesplora Aquarium and Qb-Test) by correlating neuropsychological data with clinical rating scales and a personality-based measure of impulsivity.

## 2. Materials and Methods

### 2.1. Participants

The clinical sample was recruited from unselected adult patients presenting to the ADHD outpatient clinic of the Department of Psychiatry, Psychosomatic Medicine, and Psychotherapy, University Hospital Frankfurt, for diagnostic assessment of possible ADHD between May and November 2019. Among the 29 participants (Table 1), 23 (79.3%) were diagnosed with ADHD after the full clinical diagnostic assessment. Six patients (20.7%) did not meet the diagnostic criteria. Comorbid mental disorders were present in 17 (73.9%) of the ADHD patients, including depression (*n* = 13, 56.5%), substance abuse disorders (*n* = 3, 13.0%), personality disorders (*n* = 2, 8.7%) and delusional disorders (*n* = 1, 4.3%). Six patients (20.7%) did not show any other comorbid mental disorder apart from ADHD. The structured diagnostic interview for ADHD in adults (DIVA 2.0) [28] was used to assess the diagnostic criteria and identify the different subtypes of ADHD. Among the ADHD patients, one patient each (4.3%) was referred to as inattentive and hyperactive/impulsive, respectively, whereas 13 patients (56.5%) were classified as combined subtype. During the diagnostic process none of the patients was taking ADHD medication. Two patients received ADHD medication before the protocol. One patient was diagnosed with ADHD in childhood and received Ritalin, which was used for a short term. The second patient received treatment in 2011 and has used Medikinet for a period of 3 months. All other patients were treatment naive. Signed informed consent forms were obtained from the participants. The study was approved by the Ethics Committee of the University Hospital Frankfurt (Votum No. 425/14).

### 2.2. Procedure

The patients were seen by experienced clinicians (O.G., N.B.-K.) to assess current and former symptoms in adulthood and childhood with the use of the structured diagnostic interview for ADHD in adults (DIVA 2.0) [28]. Childhood school records were assessed. Medical history including former substance abuse, mental disorders comorbidities, somatic disorders, and family anamnesis of ADHD were assessed. Patients were interviewed with the Wender–Reimherr Interview (WRI) [29] by an independent trained interviewer (A.B.) and were asked to fill out the following self-report instruments: short form of the Wender–Utah Rating Scale (WURS-k, cut off for the diagnosis of a childhood ADHD sum score ≥30) [30], Impulsive Behavior Scale (UPPS) [31] and the General Health Questionnaire (GHG-28) [32]. Afterwards, neuropsychological testing took place. All patients were tested with two continuous performance tests: the Quantified Behavior Test [26] and Nesplora Aquarium [25]. Testing order was randomized across the sample.

### 2.3. Neuropsychology

#### 2.3.1. Nesplora Aquarium

Nesplora Aquarium is a computerized CPT based on VR [25]. Through VR optical devices participants are immersed in a virtual aquarium. Throughout the test, visual and auditory stimuli are being presented to the patients and different tasks must be performed. There is a training task at the beginning to familiarize with the stimuli and the equipment. Through the sensors in the VR-glasses head movements are measured. After the training task, which is not included in the evaluation, two tasks are to be performed (Figure 1a,b). The test lasts about 18 min and includes training items in every task, which are not considered in the evaluation. The description of the variables measured in the test can be obtained from the Appendix B (Table A1).

#### 2.3.2. Quantified Behavior Test

The Qb-Test [26] measures the cardinal symptoms of ADHD: inattention, impulsivity and hyperactivity in a computerized test lasting for about 20 min. The equipment consists of a headband with a reflector, which is captured by an infrared camera to measure motor activity during the test. As shown in Figure 1c visual stimuli are being presented, and the participant is supposed to press a button whenever there is an exact repetition of the prior stimulus. Whenever the stimuli are not repeated, participants are instructed to withhold pressing. The test provides raw scores of inattention (omission errors, reaction time, variation of reaction time during the second half of the test), impulsivity (commission errors, normalized commission errors) and hyperactivity (time active, distance, area, micro events, motion simplicity). The description of the variables can be obtained from the Appendix B (Table A1). Qb-Tech provides single variables as well as the three principal components of inattention, impulsivity, and hyperactivity, derived from a principal component analysis.

#### 2.3.3. Diagnostic Interview for ADHD in adults (DIVA)

DIVA 2.0 (Diagnostic Interview for ADHD in adults) [28] is a semi-structured interview assessing the ADHD criteria based on the fourth edition of the Diagnostic and Statistical Manual of Mental Disorders (DSM-IV) [33]. The DIVA 2.0 consists of three parts that are each applicable for both childhood (before age 12) and adulthood. The first and the second parts assess the DSM-IV criteria for the core symptom clusters of ADHD including inattention and hyperactivity/impulsivity, respectively. For each criterion and age group, specific examples are provided. The third part evaluates functional impairment caused by the ADHD symptoms in five domains (including work/education, relationships and family life, social contacts, free time/hobbies, self-confidence/self-image) specifically during adulthood and childhood. Though the DIVA 2.0 is based on DSM-IV, the diagnosis was made based on the current DSM-5, whereby it applies, that symptoms are sufficient to make the ADHD diagnose.

#### 2.3.4. Wender–Reimherr Interview (WRI)

The WRI [29] is the German adaptation of the Wender–Reimherr Adult Attention Deficits Disorders Scale (WRAADDS) [34]. The WRI is a structured interview for assessing ADHD core symptoms and other related problems in adults. The interview includes 28 questions covering seven psychopathological domains including attention deficit, hyperactivity/restlessness, temper, affective lability, emotional over-reactivity, disorganization, and impulsivity. The questions are rated on a scale from 0 “Does not apply” to 2 “Often occurs”. The WRI was used for generating a dimensional score of ADHD symptoms. The diagnosis was based on the DSM-V criteria.

#### 2.3.5. Wender–Utah Rating Scale (WURS-k)

The WURS-k [30] is a self-report measure that is used to retrospectively evaluate childhood ADHD symptoms. We used the short form comprising 25 items for discriminating patients with ADHD versus controls. The items are rated on a 5-point response scale ranging from 0 “Not at all” to 4 “Very much”. A total score of 30 is considered as a cutoff to diagnose ADHD in childhood [35].

#### 2.3.6. UPPS Impulsive Behavior Scale (UPPS)

The UPPS [31] is a self-report measure consisting of 45 items that are rated on a 4-point Likert scale ranging from 1 (strongly agree) to 4 (strongly disagree). The UPPS includes four subscales: urgency, lack of premeditation, lack of perseverance, and sensation seeking. In a psychometric study of the German adaptation of the UPPS in a German-speaking sample [36], exploratory and confirmatory factor analyses showed a four-factor structure similar to the results in the original study. The four subscales showed a very good internal consistency with Cronbach’s alpha ranging from 0.80 to 0.85 [36].

#### 2.3.7. General Health Questionnaire-28 (GHQ-28)

The GHQ-28 [32] was developed to assess emotional distress and consists of 28 items, with four subscales including somatic symptoms, anxiety/insomnia, social dysfunction, and severe depression. The items can be scored from 0 to 3.

### 2.4. Data Analysis

To enable a comparison between the principal components of the Qb-Test and equivalent parameters of Nesplora Aquarium, a principal component analysis was performed on the variables of Nesplora Aquarium. Based on the description of the measures provided by Nesplora Aquarium [25], the 15 named variables (described in Appendix B, Table A1) were included. A varimax rotation was used as the factors were expected to be independent. Item loadings below 0.30 were not considered. The scree plot can be found in the Appendix A (Figure A1). The correlation matrix was nonpositive definite, which means that some of the eigenvalues are not positive values. In this case the Kaiser-Meyer-Olkin measure and Bartlett test cannot be calculated. Eigenvalues were calculated for each factor in the data, whereas five factors had eigenvalues over Kaiser’s criterion of 1. The initial eigenvalues explained in sum 89.85% of variance. The factor loadings after rotation can be obtained from Table 2.

The variables that highly load on factor 1 (“Sum of distance (mean)”, “Movement in the yaw shaft (mean)”, “Movement in the pitch shaft (mean)”, “Movement in the roll shaft (mean)”) are measures of movement, implicating that factor 1 represents “Motor activity”. Factor 1 explains 36.62% of variance. The second factor “Inattention” is formed by 5 variables that loaded highest on this factor (“Total omission errors (*n*)”, “Dual task correct answers (*n*)”, “Correct answers reaction time (SD) (ms)”, “Commission errors reaction time (SD) (ms)”, “Perseverative errors (*n*)”) and explains 22.57% variance. The three variables that cluster on the third factor “Switching” are represented by the reaction time of hits and the total number of hits (“Switching reaction time (mean)”, “Switching correct answers (*n*)”) as well as by the “Discrepancy of correct answers (*n*)”. 13.04% variance can be explained by factor 3. Factor 4 “Reaction time” is represented by “Commission errors reaction time (mean) (ms)” and “Correct answers reaction time (mean) (ms)” and explains 10.53% variance. The last factor 5 is formed by only one variable namely “Total commission errors (*n*)”. The 7.08% variance is explained by factor 5 “Impulsivity”. The factors were included in the correlational analyses with the principal components of the Qb-test. To test the comparability of the two CPTs, correlational analyses were performed. Before, differences in means and standard deviations were calculated (Table 1). We then correlated the five principal components of Nesplora Aquarium derived from the principal component analysis with the three principal components of the Qb-Test, which have previously been calculated and provided by Qbtech (Table 3). To enable a more detailed insight in the particular relationships between the single variables, all variables of Nesplora Aquarium and the Qb-Test, that formed the principal components, were correlated. In a last step correlational analyses were performed to assess the compatibility of the two CPTs with clinical measures. Firstly, the principal components of both CPTs and secondly all variables provided by the CPTs were correlated with clinical rating scales (WRI, WURS-k, GHQ-28) and the personality-based measure (UPPS). The analyses were run with IBM SPSS Statistics (IBM Corp. Released 2017. IBM SPSS Statistics for Macintosh, Version 25.0. Armonk, NY: IBM Corp.) [37]. Although this was an exploratory study on a highly specific, small sample of ADHD patients, we additionally corrected for multiple testing when necessary. Because of the high collinearity of our variables, we calculated the number of independent tests as done previously [38,39]. We calculated the principal components explaining 90% variance for each table to estimate the effective number of independent tests performed in our correlational analyses (Table 4, Table 5 and Table 6). Afterwards, the Bonferroni correction formula was used to calculate the adjusted significance levels, as proposed [39]: 0.05/8 = ^†^
*p* = 0.006 (Table 4); 0.05/9 = ^†^
*p* = 0.006 (Table 5); 0.05/10 = ^†^
*p* = 0.005 (Table 6).

## 3. Results

Partial correlations (controlling for sex, age, and order of test administration) were carried out between the principal components of Nesplora Aquarium and the Qb-Test. A significant positive correlation was found between the inattention factors of Nesplora Aquarium and the Qb-Test (*r* = 0.49, *p* = 0.01) (Table 3).

To further specify the relation between the two continuous performance tests, partial correlations (controllin1.g for sex, age, and order of test administration) between the underlying variables of the principal components of Nesplora Aquarium and the Qb-Test were performed. Highly significant correlations were found especially among the inattention variables of the Qb-Test, correlating with variables of Nesplora Aquarium (Table 4): Omission errors as well as correct answers reaction time (SD), indicating sustained attention or fatigability, positively correlated between both tests (*r* = 0.58 and 0.52, respectively, *p* = 0.002 and 0.006, respectively). Omission errors in the Qb-Test negatively correlated with working memory, measured by “dual task correct answers” in Nesplora Aquarium (*r* = −0.53, *p* = 0.006), whereby low scores indicate deficits in working memory. Furthermore, omission errors (Qb-Test) were associated with movement, measured by Nesplora Aquarium (“movement in the roll shaft”) (*r* = 0.53, *p* = 0.006). All associations described were highly significant under the corrected significance value of *p* < 0.006 after adjusting for multiple testing.

In a next step neuropsychological data was analyzed regarding the compatibility with clinical measures. Table 5 shows partial correlations (controlling for sex, age, and order of test administration) between the principal components of Nesplora Aquarium and the Qb-Test with the WRI, WURS-k, UPPS, and GHQ-28. Among all clinical parameters, the WRI was found to be related mostly to neuropsychological data, whereby two correlations were highly significant: affective lability in the WRI negatively correlated with impulsivity in Nesplora Aquarium (*r* = −0.62, *p* = 0.008), and overactivity in the WRI showed a positive correlation with hyperactivity in the Qb-Test (*r* = 0.71, *p* = 0.001). A positive significant correlation was found between WURS-k and hyperactivity in the Qb-Test (*r* = 0.50, *p* = 0.04). The urgency subscale of the UPPS negatively correlated with impulsivity in the Qb-Test (*r* = −0.49, *p* = 0.046). No significant correlations were found between the GHQ-28 and neuropsychological measures. The described positive correlation between overactivity in the WRI and hyperactivity in the Qb-Test showed to be highly significant after correction for multiple testing (*p* < 0.006).

In a closer examination of the relations between neuropsychological data and clinical measures, partial correlations (controlling for sex, age, and order of test administration) between the underlying variables of the principal components of Nesplora Aquarium and the Qb-Test with the WRI, WURS-k, UPPS, and GHQ-28 were carried out (Table 6). In line with the previous results, the WRI showed the most significant correlations with the continuous performance tests: Overactivity in the WRI negatively associated with discrepancy of correct answers (*r* = −0.63, *p* = 0.007) in Nesplora Aquarium while it positively correlated with area, which measures movement in the Qb-Test (*r* = 0.63, *p* = 0.007). Attention disorder in the WRI correlated positively with motion simplicity in the Qb-Test (0.64, *p* = 0.005). The correlation revealed to be highly significant after correction of multiple testing (*p* < 0.005). The total WRI score was positively associated with movement in Nesplora Aquarium (*r* = 0.61, *p* = 0.009).

## 4. Discussion

The current study aimed to compare Nesplora Aquarium and the Qb-Test in a within-sample comparison of adult patients presenting to our ADHD outpatient clinic and related test measures to clinical scores. The ecological validity of the two CPTs (i.e., Nesplora Aquarium and Qb-Test) was assessed regarding the correlation with clinical rating scales (WRI, WURS-k, GHQ-28) and a personality-based measure (UPPS). The overall comparability of the tests was limited. Despite the congruent measure of inattention in both tests, no correlation with clinical features was found. While hyperactivity positively associated with current ADHD symptoms in both tests, in the Qb-Test it additionally reflected childhood ADHD symptoms. Impulsivity was shown to be represented independently by both tests and poorly associated with clinical measures and UPPS. We did not only investigate a previously described component structure of the Qb-test, but we also calculated the varimax-rotated principal component analysis of the Nesplora Aquarium test to better compare basic components of both tests.

The factor inattention was found to positively associate between the two CPTs in a moderate way. A closer look at the variables, which define the factor inattention, confirms this finding. Omission errors and the variation of reaction time, indicating consistency of attention and fatigability, were correlated in both tests. Not surprisingly, inattention is the most basic bottleneck of both tests. While both tests seem to be well designed in assessing the feature, inattention in the tests does not have strong relations with clinical parameters.

Nesplora Aquarium provides the additional measure of working memory, defined by the parallel processing of two sensory modalities during test performance. Working memory was found to be negatively associated with omission errors in the Qb-Test, indicating that inattention is associated with significant impairment in the working memory.

Perseverative errors are another new variable of Nesplora Aquarium measuring deficits in cognitive flexibility. It was found to positively correlate with omission errors in the Qb-Test. The present findings indicate that inattention is diversely captured by CPTs, especially by Nesplora Aquarium.

We did not detect other overlaps between major components of the two tests. The correlational analyses of the underlying variables measuring hyperactivity and impulsivity did not present consistent findings, respectively. The variables of movement among the two tests did not correlate.

In the present study, 79.3% of the patients that presented to the outpatient clinic were finally diagnosed with ADHD. Apart from assessing the diagnostic accuracy of CPTs as done by a range of authors [16,17], our study allows a more complex look at the various features of ADHD, captured by clinical rating scales and a personality-based measure, and how well they are represented through neuropsychological testing. For this purpose, we evaluated the relationship of the CPTs with clinical measures including WRI for assessing ADHD symptoms in adulthood, WURS-k for a retrospective assessment of ADHD symptoms during childhood, UPPS for measuring impulsivity, and GHQ-28 for identifying relationships with short-term psychiatric disorders. In the following the results concerning the ecological validity of the CPTs will be reported for inattention, hyperactivity and impulsivity, respectively.

The factor inattention in both Nesplora Aquarium and the Qb-Test did not show any relationship with clinical rating scales or UPPS. The same applies to the factor reaction time in Nesplora Aquarium. The lack of clinical relevance of inattention is surprising, since inattention is the most consistent factor across paradigms. Also, the strong and reasonable correlations of the underlying variables measuring inattention in both tests gave reason to expect significant clinical correlates. In contrast to our findings, previous studies found low attention performance (omission errors and number of correct answers) measured by Nesplora Aquarium to positively correlate with current and retrospective ADHD symptoms [14]. In a study on the role of objective measures in assessing ADHD symptoms in children and adults [40], the inattention factor of the Qb-Test positively correlated with the inattention subscale of the Conner’s ADHD rating scales-observer ratings (CAARS-O) in adults. Nevertheless, the similar metric profile of omission errors, commission errors and hit reaction time in VR and non-VR tests is not able to affect the ecological validity just by adapting the test environment to a more real-world situation [16].

Measures of movement in the Qb-Test were found to associate with commission errors in Nesplora Aquarium, a measure of impulsivity. Variables indicating movement in Nesplora Aquarium most likely related to omission errors in the Qb-Test, a measure of inattention. A possible explanation for the lack of correlation between the activity variables of the two CPTs is that they measure movement in two different ways. In the Qb-Test, the participant’s movements are recorded by using an infrared system tracking a reflective indicator located on the headband participants wear [41]. However, in Nesplora Aquarium, head movements of the participants are recorded by sensors placed in the glasses [27]. Movement might be a factor, which is not independent of the task but closely related to cognitive demands. Putting constraints to head movement (like balancing VR glasses) might impact on cognitive capabilities itself [42].

Although hyperactivity did not show significant correlation in the CPTs, a positive association with overactivity, a subscale of the WRI, was found in both tests. These findings apply for the hyperactivity factor in both CPTs as well as the underlying variables of hyperactivity in Nesplora Aquarium and the Qb-Test. These results are supported by the finding that ADHD patients show significant higher activity than patients without ADHD [19]. While the hyperactivity factors of both tests are independent, they nevertheless seem to be related to one of the most valid clinical symptom components. The additional correlations of hyperactivity with several other subscales of the WRI (overactivity, affective lability, impulsiveness for Nesplora Aquarium; overactivity and temperament for Qb-Test) indicate that hyperactivity adequately represents current symptoms of ADHD. Interestingly, affective lability, an often dismissed core symptom of adult ADHD was indexed by Nesplora Aquarium’s movement parameters while it was not detected by Qb-Test’s variables. The fact that Nesplora Aquarium is the more demanding and complex test, which can lead to frustration, could explain this finding. Accordingly, we assume that patients with higher affective lability show more motor activity during such tests involving a high frustration potential. Besides, affective lability is ignored by the DSM-V criteria for adult ADHD although it represents a highly important negative part in daily life of patients.

Moreover, childhood ADHD symptoms captured by the WURS-k are linked to hyperactivity in the Qb-Test. In line with our findings, hyperactivity was found to be associated with current and retrospective ADHD symptoms in another study [43]. The findings suggest that hyperactivity measured in neuropsychological testing represents the most accurate correlate of ADHD symptoms.

Impulsive decision-making might be a basic feature for measuring ADHD. However, Qb-Test’s factor impulsivity, which is mainly driven by commission errors in the Qb-Test did not correlate with measures of impulsivity in Nesplora Aquarium at all, questioning whether there is a homogenous impulsivity construct across neuropsychological tests and clinical symptoms. Impulsive behavior can be differentiated into several distinct and heterogenous subtypes [44]. Moreover, impulsivity is not consistent throughout patient populations, but takes various forms according to different psychiatric disorders [45]. In the current study, impulsivity measured by the Qb-Test negatively correlated with urgency, a subscale of the UPPS. Impulsivity in Nesplora Aquarium did not show any association with the UPPS, but was negatively linked to affective lability, a subscale of the WRI. ADHD patients with high affective lability might be more afraid to make mistakes and withhold pressing rather than falsely reacting to a non-target stimulus. The positive correlation between affective lability and commission errors reaction time (SD) additionally emphasizes the variance present in both affective lability, by its definition, and commission errors reaction time. This leads to the assumption that high affective lability in ADHD patients is associated with great variability in pressing or withholding, expressed by higher motor activity driven by frustration, as mentioned above, respectively.

While sensation seeking, a subscale of the UPPS, is discussed as one of the most prominent traits of ADHD, none of the test measures of Qb-Test or Nesplora Aquarium captured this trait in patients. The lacking correlation of the total UPPS score with any of the test parameters of both CPTs emphasizes the idea that ADHD symptoms are not simply about the trait impulsivity. The CPTs showed no significant correlation with the impulsiveness subscale of the WRI. Neither subscales of the UPPS nor impulsiveness in the WRI related to commission errors in any of the CPTs. The independence of the impulsivity factors of both CPTs as well as their poor association with measures of impulsivity through clinical measures and UPPS underlines the difficulty to narrow impulsivity to a uniform construct. Our findings support the idea that behavioral components and self-reported impulsivity are largely independent [46].

The findings concerning the GHQ-28 indicate that both CPTs seem to be very specific for ADHD symptoms, as the GHQ-28 captures more general emotional distress symptoms and was almost not correlated with test measurements.

Some limitations of the study should be acknowledged. The small sample size is a limiting factor in generalizing the results on a clinical level. Making recommendations about the superiority of one of the tests in predicting specific ADHD-related symptoms would require larger study samples including control groups. However, for a within-subject repeated measure study, the sample size is comparable to that of other method comparison studies. As the sample mainly consisted of ADHD patients including a control group would be of interest. 73.9% of the patients showed comorbid disorders including depression, substance abuse disorders, personality disorders and delusional disorders. Exploring the results, taking into account the comorbidities, would be of interest for future research. However, in our study the comorbidities were not evenly distributed and our sample was small, so we were not able to study them systematically. Though this study was an exploratory analysis, correction for multiple testing was applied when necessary. Most of the highly significant correlations found proved to be significant after the applied corrected significance threshold. Further studies are needed to confirm our findings in a larger data set. Apart from these limitations, our study has some unique features: We used two commercially available continuous performance tests to measure the core symptoms of ADHD. Both tests are outstanding in terms of their technology: They make it possible to capture aspects of ADHD, which are ignored by classical paper-and-pencil-tests. The Qb-Test measures hyperactivity with the use of an infrared camera and a reflector attached to a headband to record movements during the test. During the test of Nesplora Aquarium, participants are immersed in a virtual aquarium, while motor activity is measured through VR optical devices equipped with sensors and headphones. We evaluated Nesplora Aquarium and the Qb-Test in the diagnostic workflow with patients of our adult ADHD outpatient clinic, providing highly specific results. Additionally, the factor structure of Nesplora Aquarium has to our knowledge not yet been examined by other studies and represents a new finding in the current paper.

## 5. Conclusions

Taken together, the current study enables a good comparison of the two CPTs and reveals detailed information on their compatibility with clinical measures. Inattention, mainly driven by test performance, is a basic factor, which overlaps between the two tests in each subject. As it fails to predict clinical parameters, this highlights the previously described problem in linking subjective and objective methods in diagnosing ADHD [19]. A very interesting test parameter is hyperactivity, which reflects current ADHD symptoms in both tests, while in the Qb-Test it is additionally associated with childhood ADHD symptoms. As there are only few findings about the compatibility of Nesplora Aquarium with clinical measures of current and childhood ADHD symptoms at this time, this issue should be addressed by further studies. Not surprisingly, both tests are linked together by the component inattention. However, as this measurement of basic test performance does not seem to have a clinical representation, we would currently not use either test as a step in the diagnostic workflow. The current findings question the simple use of so-called objective measures in clinical ADHD diagnosis because of questionable validity. The findings emphasize the need for a new approach in diagnosis in the sense of precision medicine, going beyond conventional classification systems to enable suitable individual solutions for diagnosis and treatment [22]. While Kapur [47] asked for cognitive tests that distinguish a broad clinical phenotype into distinct treatment-relevant subgroups, our findings illustrate that such cognitive tests are hard to establish. Though continuous performance tests are intended to improve diagnosis, the current test’s validity is not clearly established. For example, impulsivity measures in both tests do not correlate with clinical parameters related to ADHD impulsivity. Movement-related parameters during complex cognitive tests show some promise and should be validated in future research in deeply phenotyped samples.

## Figures and Tables

**Figure 1 brainsci-11-00036-f001:**
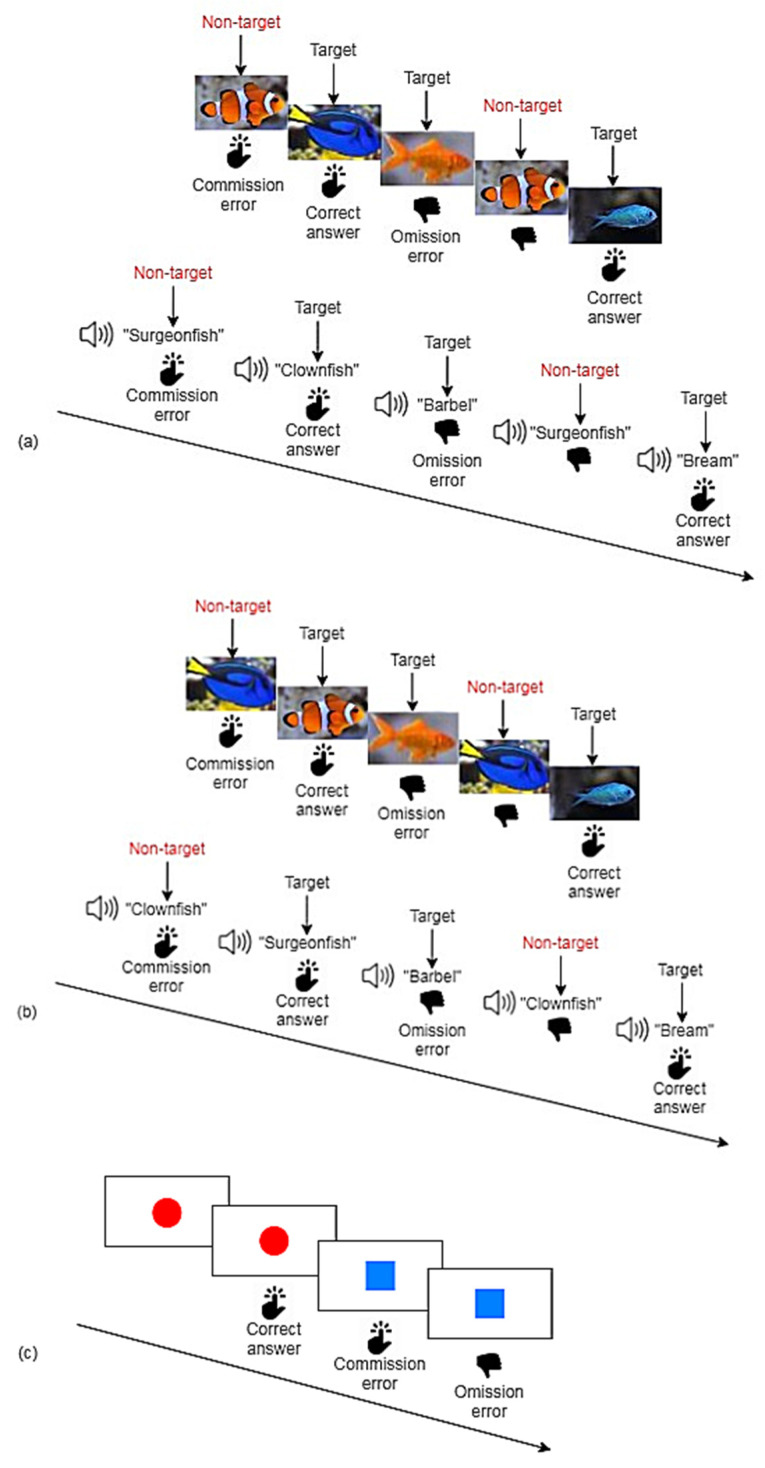
Schematic diagrams of Nesplora Aquarium and Qb-Test. (**a**) In the first step of Nesplora Aquarium, the participant’s task is to press the button in response to all visual stimuli except for clownfish and to all auditory stimuli except for surgeonfish. (**b**) In the second step of Nesplora Aquarium, the participant is asked to press the button while observing all stimuli except for surgeonfish and while hearing all stimuli except for clownfish (inversed task). (**c**) In the Qb-Test, the participant’s task is to press a button whenever an exact repetition of the prior stimulus occurs.

**Table 1 brainsci-11-00036-t001:** Sample and test characteristics.

*n* Demographics	29
Age (M (years)) ± SD	35.24 ± 11.05
Sex	
Males	10 (34.5%)
Females	19 (65.5%)
Diagnosis	
ADHD	23 (79.3%)
Non-ADHD	6 (20.7%)
Comorbid disorders among ADHD patients	17 (73.9%)
Depression	13 (56.5%)
Substance abuse disorders	3 (13.0%)
Personality disorders	2 (8.7%)
Delusional disorders	1 (4.3%)
ADHD testing	
DIVA-ADHD subtypes among ADHD patients	
Inattentive	1 (4.3%)
Hyperactive/Impulsive	1 (4.3%)
Combined	13 (56.5%)
WRI (*n* = 27, mean, (±SD))	
Attention disorder	6.81 (±3.03)
Overactivity	3.7 (±1.82)
Temperament	2.89 (±2.29)
Affective lability	3.85 (±2.48)
Stress intolerance	3.74 (±2.19)
Disorganization	6.93 (±2.97)
Impulsiveness	5.41 (±2.1)
Total	4.76 (±1.7)
WURS-k total (*n* = 26 mean, (±SD))	36.81 (±11.23)
Psychological Questionnaires	
GHQ-28 (*n* = 25 mean, (±SD))	1.17 (±0.47)
Somatic symptoms	1.17 (±0.52)
Anxiety/insomnia	1.35 (±0.61)
Social dysfunction	1.39 (±0.53)
Severe depression	0.77 (±0.6)
UPPS (*n* = 22 mean, (±SD))	
Urgency	2.53 (±0.39)
Lack of premeditation	2.53 (±0.48)
Lack of perseverance	2.54 (±0.26)
Sensation seeking	2.56 (±0.33)
Total	2.54 (±0.15)
Neuropsychological testing	
Nesplora principal components (*n* = 29 mean, (±SD))	
Motor activity	0.00 (±1.00)
Inattention	0.00 (±1.00)
Switching	0.00 (±1.00)
Reaction time	0.00 (±1.00)
Impulsivity	0.00 (±1.00)
Qb-Test principal components (*n* = 29 mean, (±SD))	
Activity	1.95 (±1.19)
Impulsivity	1.08 (±1.01)
Inattention	0.94 (±1.17)
Nesplora variables (*n* = 29 mean, (±SD))	
Perseverative errors (*n*)	16.66 (±6.78)
Switching reaction time (mean) (ms)	−8894.12 (±2611.34)
Switching correct answers (*n*)	−16.67 (±2.72)
Total commission errors (*n*)	18.17 (±7.28)
Commission errors reaction time (mean) (ms)	747.97 (±189.27)
Commission errors reaction time (SD) (ms)	133,201.5 (±82,565.08)
Discrepancy of correct answers (*n*)	−0.121 (±3.18)
Correct answers reaction time (mean) (ms)	953.83 (±119.48)
Correct answers reaction time (SD) (ms)	98,724.49 (±35,981.23)
Sum of distance (mean)	0.2914 (±0.26)
Dual task correct answers (*n*)	119.64 (±10.34)
Total omission errors (*n*)	22.55 (±19.5)
Movement in the pitch shaft (mean)	0.10 (±0.07)
Movement in the roll shaft (mean)	0.08 (±0.08)
Movement in the yaw shaft (mean)	0.11 (±0.12)
Qb-Test variables (*n* = 29 mean, (±SD))	
Time active (>1 cm/s)	23.62 (±16.91)
Distance (m)	12.71 (±10.44)
Area (cm^2^)	49.10 (±39.04)
Micro events (>1 mm)	6301.9 (±4149.57)
Motion simplicity (*n*)	44.24 (±12.34)
Total omission errors (*n*)	12.39 (±13.91)
Total commission errors (*n*)	1.74 (±1.67)
Correct answers reaction time (mean)	609.9 (±109.69)
Correct answers reaction time (SD)	177.52 (±61.35)

*n* = size of sample, M = mean, SD = standard deviation, ms = millisecond, cm = centimeter, s = second, m = meter, mm = millimeter, cm^2^ = square centimeter. ADHD = attention deficit hyperactivity disorder, DIVA = structured diagnostic interview for ADHD in adults, WURS-k = short form of Wender-Utah Rating Scale.

**Table 2 brainsci-11-00036-t002:** Rotated Factor Loadings.

	Rotated Factor Loadings				
	Motor Activity	Inattention	Switching	Reaction Time	Impulsivity
Sum of distance (mean)	0.98				
Movement in the yaw shaft (mean)	0.96				
Movement in the pitch shaft (mean)	0.95				
Movement in the roll shaft (mean)	0.93				
Total omission errors (*n*)		0.95			
Dual task correct answers (*n*)		−0.92			
Correct answers reaction time (SD) (ms)		0.92			
Commission errors reaction time (SD) (ms)		0.73		0.49	
Perseverative errors (*n*)		0.67	0.31		0.48
Switching reaction time (mean) (ms)			0.91		
Switching correct answers (*n*)		0.31	0.87		
Discrepancy of correct answers (*n*)		0.41	−0.73		
Commission errors reaction time (mean) (ms)				0.89	
Correct answers reaction time (mean) (ms)				0.80	
Total commission errors (*n*)					0.95

*n* = number of, ms = millisecond.

**Table 3 brainsci-11-00036-t003:** Correlations between the principal components of Nesplora Aquarium and the Qb-Test.

Qb-Test			
	Hyperactivity	Impulsivity	Inattention
Nesplora Aquarium	*r*	*r*	*r*
Motor activity	0.23	−0.29	0.28
Inattention	0.26	−0.02	0.49 *
Switching	0.26	0.23	−0.25
Reaction time	−0.22	−0.21	−0.35
Impulsivity	0.06	0.04	−0.007

* *p* < 0.05.

**Table 4 brainsci-11-00036-t004:** Correlations between the underlying variables of the principal components of Nesplora Aquarium and the Qb-Test.

	Qb-Test									
	Hyperactivity					Inattention			Impulsivity	
	Time Active (>1 cm/s)	Distance (m)	Area (cm^2^)	Micro Events (>1 mm)	Motion Simplicity (*n*)	Omission Errors (*n*)	Correct Answers Reaction Time (mean)	Correct Answers Reaction Time (SD)	Commission Errors (*n*)	Normalized Commission Errors (*n*)
Nesplora Aquarium										
Perseveration errors (*n*)	0.12	0.14	0.23	0.15	0.40 *	0.41 *	−0.05	0.22	−0.21	−0.20
Switching reaction time (mean) (ms)	0.18	0.28	0.29	0.23	0.03	0.03	−0.36	−0.009	0.14	0.13
Switching correct answers (*n*)	0.26	0.26	0.28	0.28	0.19	0.11	−0.27	0.02	0.15	0.16
Total commission errors (*n*)	0.18	0.10	0.16	0.16	0.27	0.05	−0.08	0.12	0.14	0.14
Total omission errors (*n*)	0.08	0.18	0.24	0.13	0.39	0.58 **^,†^	0.27	0.50 *	−0.03	0.03
Commission errors reaction time (mean) (ms)	−0.38	−0.41 *	−0.41 *	−0.41 *	−0.38	−0.02	−0.34	−0.28	−0.20	−0.19
Commission errors reaction time (SD) (ms)	−0.03	0.03	0.13	0.02	0.36	0.33	0.07	0.17	−0.23	−0.21
Correct answers reaction time (mean) (ms)	−0.11	−0.11	−0.09	−0.11	−0.02	−0.04	−0.12	−0.19	−0.25	−0.26
Correct answers reaction time (SD) (ms)	0.03	0.09	0.17	0.06	0.43 *	0.48 *	0.38	0.52 **^,†^	0.15	0.19
Discrepancy of correct answers (*n*)	−0.36	−0.27	−0.32	−0.34	−0.19	0.18	0.42 *	0.35	0.23	0.26
Dual task correct answers (*n*)	−0.13	−0.19	−0.27	−0.17	−0.44 *	−0.53 **^,†^	−0.21	−0.48 *	−0.02	−0.08
Sum of distance (mean)	0.10	0.08	0.17	0.12	0.38	0.46 *	0.14	0.20	−0.08	−0.05
Movement in the pitch shaft (mean)	0.11	0.07	0.15	0.13	0.35	0.35	0.01	0.08	−0.10	−0.06
Movement in the roll shaft (mean)	0.08	0.04	0.13	0.09	0.36	0.53 **^,†^	0.17	0.18	−0.20	−0.17
Movement in the yaw shaft (mean)	0.09	0.11	0.20	0.12	0.38	0.45 *	0.18	0.27	0.02	0.06

*n* = number of, SD = standard deviation, ms = millisecond, cm = centimeter, s = second, m = meter, mm = millimeter, cm^2^ = square centimeter. ** *p* < 0.01. * *p* < 0.05. † *p* < 0.006.

**Table 5 brainsci-11-00036-t005:** Correlations between the principal components of Nesplora Aquarium and the Qb-Test with WRI, WURS-k, UPPS, and GHQ-28.

	WRI										
	Attention Disorder	Overactivity	Temperament	Affective Lability	Stress Intolerance	Disorganization	Impulsiveness	TOTAL		Total WURS-k	
Nesplora Aquarium											
Motor activity	0.25	0.52 *	0.36	0.61 *	0.32	0.30	0.53 *	0.58 *		−0.08	
Inattention	0.48	0.08	−0.17	0.22	0.40	0.30	0.30	0.34		0.33	
Switching	0.41	0.56 *	0.37	0.39	−0.02	0.08	−0.03	0.34		0.06	
Reaction time	−0.36	0.12	0.26	0.03	0.008	−0.15	−0.18	−0.07		0.10	
Impulsivity	−0.14	−0.32	−0.46	−0.62 **	−0.34	−0.36	−0.17	−0.50 *		−0.32	
QB-Test											
Hyperactivity	0.45	0.71 **^,†^	0.52 *	0.47	0.20	−0.02	0.47	0.54 *		0.50 *	
Impulsivity	0.34	−0.22	−0.46	−0.21	0.08	0.37	−0.08	−0.003		−0.09	
Inattention	0.28	−0.29	−0.42	0.02	0.37	0.20	0.13	0.09		0.16	
	UPPS						GHQ−28				
	Urgency	Lack of Premeditation	Lack of Perseverance	Sensation Seeking	Total		Somatic Symptoms	Anxiety/Insomnia	Social Dysfunction	Depression	Total
Nesplora Aquarium											
Motor activity	0.15	−0.09	0.08	0.16	0.14		−0.14	0.03	0.06	−0.001	−0.01
Inattention	−0.37	0.15	0.07	0.02	−0.05		0.20	0.18	0.12	−0.01	0.15
Switching	0.32	−0.45	−0.41	0.17	−0.24		0.12	0.23	0.42	−0.11	0.21
Reaction time	0.04	−0.18	0.19	0.03	−0.03		0.21	0.41	0.23	0.39	0.39
Impulsivity	0.08	0.27	−0.05	−0.46	−0.03		0.13	−0.19	−0.02	0.33	0.06
QB-Test											
Hyperactivity	0.44	−0.44	−0.27	0.25	−0.06		0.28	0.44	0.46	0.15	0.42
Impulsivity	−0.49 *	0.04	0.07	−0.13	−0.28		−0.20	−0.28	−0.23	−0.07	−0.24
Inattention	−0.44	0.48	0.34	−0.12	0.19		−0.32	−0.19	−0.13	−0.31	−0.29

WRI = Wender–Reimherr Interview, WURS-k = Wender–Utah Rating Scale, UPPS = UPPS Impulsive Behavior Scale, GHQ-28 = General Health Questionnaire-28. ** *p* < 0.01. * *p* < 0.05. † *p* < 0.006.

**Table 6 brainsci-11-00036-t006:** Correlations of the underlying variables of the principal components of Nesplora Aquarium and the Qb-Test with WRI, WURS-k, UPPS, and GHQ-28.

		WRI								
		Attention Disorder	Overactivity	Temperament	Affective Lability	Stress Intolerance	Disorganization	Impulsiveness	Total	Total WURS-k
	Nesplora Aquarium									
Perseveration errors (*n*)		0.50 *	0.52 *	0.11	0.33	0.21	0.09	0.14	0.38	0.29
Switching reaction time (mean) (ms)		0.50 *	0.55 *	0.25	0.43	0.03	0.19	−0.03	0.39	0.009
Switching correct answers (*n*)		0.48	0.45	0.19	0.27	0.02	0.13	0.11	0.33	0.1
Total commission errors (*n*)		0.11	−0.11	−0.39	−0.38	−0.17	−0.21	0.08	−0.22	−0.29
Total omission errors (*n*)		0.50 *	0.12	−0.08	0.33	0.42	0.37	0.37	0.43	0.25
Commission errors reaction time (mean) (ms)		−0.50 *	−0.20	0.01	−0.22	−0.08	−0.16	−0.33	−0.30	−0.10
Commission errors reaction time (SD) (ms)		0.21	0.37	0.29	0.50 *	0.48	0.17	0.28	0.46	0.35
Correct answers reaction time (mean) (ms)		−0.29	0.11	0.19	0.01	0.02	−0.08	−0.10	−0.04	0.26
Correct answers reaction time (SD) (ms)		0.35	−0.05	−0.26	−0.005	0.26	0.19	0.18	0.15	0.26
Discrepancy of correct answers (*n*)		−0.22	−0.63 **	−0.60 *	−0.43	0.04	0.08	−0.04	−0.34	−0.02
Dual task correct answers (*n*)		−0.56 *	−0.09	0.21	−0.22	−0.38	−0.32	−0.42	−0.38	−0.17
Sum of distance (mean)		0.42	0.53 *	0.24	0.58 *	0.33	0.36	0.53 *	0.61 *	−0.07
Movement in the pitch shaft (mean)		0.36	0.43	0.28	0.53 *	0.28	0.34	0.54 *	0.56 *	−0.18
Movement in the roll shaft (mean)		0.38	0.60 *	0.32	0.54 *	0.24	0.18	0.58 *	0.56 *	0.06
Movement in the yaw shaft (mean)		0.44	0.53 *	0.17	0.59 *	0.39	0.42	0.46	0.61 **	−0.05
	Qb−Test									
Time active (>1 cm/s)		0.27	0.55 *	0.42	0.18	−0.10	−0.04	0.26	0.29	0.45
Distance (m)		0.51 *	0.58 *	0.42	0.33	0.06	0.19	0.3	0.48	0.58 *
Area (cm^2^)		0.60 *	0.63 **	0.44	0.38	0.09	0.17	0.31	0.52 *	0.55 *
Micro events (>1 mm)		0.4	0.61 *	0.48	0.29	−0.02	0.06	0.32	0.41	0.53 *
Motion simplicity (*n*)		0.64 **^,†^	0.45	0.44	0.41	0.25	−0.07	0.39	0.49 *	0.38
Omission errors (*n*)		0.28	0.07	−0.27	0.23	0.47	0.22	0.35	0.29	0.37
Correct answers reaction time (mean) (ms)		0.02	−0.37	−0.38	−0.13	0.22	−0.02	0.22	−0.08	0.24
Correct answers reaction time (SD) (ms)		0.3	−0.22	−0.35	0.04	0.37	0.35	0.11	0.15	0.26
Commission errors (*n*)		0.26	−0.23	−0.34	−0.18	0.07	0.32	0.06	0.02	0.03
Normalized commission errors(*n*)		0.26	−0.26	−0.39	−0.17	0.11	0.34	0.12	0.03	0.04
	UPPS					GHQ-28				
	Urgency	Lack of Premeditation	lack of Perseverance	Sensation Seeking	Total	Somatic Symptoms	Anxiety/Insomnia	Social Dysfunction	Depression	Total
Nesplora Aquarium										
Switching correct answers (*n*)	0.16	−0.32	−0.34	0.08	−0.25	0.07	0.22	0.39	0.09	0.24
Total commission errors (*n*)	0.06	0.25	−0.12	−0.40	−0.05	0.13	−0.17	0.02	0.26	0.06
Total omission errors (*n*)	−0.35	0.12	−0.05	−0.01	−0.13	0.18	0.2	0.1	−0.08	0.13
Commission errors reaction time (mean) (ms)	−0.10	0.02	0.33	−0.08	0.04	0.12	0.22	0.12	0.43	0.27
Commission errors reaction time (SD) (ms)	−0.18	−0.07	0.21	0.17	0.02	0.23	0.46	0.3	0.13	0.35
Correct answers reaction time (mean) (ms)	−0.01	−0.19	0.13	0.08	−0.06	0.21	0.38	0.09	0.27	0.3
Correct answers reaction time (SD) (ms)	−0.38	0.18	0.16	0.05	0.02	0.22	0.14	0.17	0.23	0.23
Discrepancy of correct answers (*n*)	−0.41	0.53 *	0.47	−0.11	0.3	−0.24	−0.43	−0.51 *	−0.08	−0.39
Dual task correct answers (*n*)	0.34	−0.21	0.09	0.15	0.15	−0.24	−0.15	−0.11	0.001	−0.16
Sum of distance (mean)	0.13	−0.06	0.04	0.15	0.12	−0.15	−0.04	0.08	−0.06	−0.05
Movement in the pitch shaft (mean)	0.11	−0.02	−0.02	0.1	0.09	−0.08	−0.007	0.09	0.03	0.01
Movement in the roll shaft (mean)	0.33	−0.11	−0.06	0.16	0.16	−0.006	0.06	0.12	0.05	0.07
Movement in the yaw shaft (mean)	0.02	−0.08	0.13	0.17	0.1	−0.24	−0.09	0.05	−0.15	−0.13
Qb−Test										
Time active (>1 cm/s)	0.50 *	−0.51 *	−0.39	0.28	−0.11	0.31	0.36	0.3	0.21	0.37
Distance (m)	0.43	−0.42	−0.30	0.38	0.01	0.1	0.22	0.09	−0.22	0.07
Area (cm^2^)	0.43	−0.40	−0.29	0.37	0.02	0.14	0.27	0.09	−0.20	0.1
Micro events (>1 mm)	0.52 *	−0.49 *	−0.36	0.34	−0.04	0.23	0.33	0.23	0.04	0.26
Motion simplicity (*n*)	0.26	−0.11	−0.22	0.13	0.05	0.3	0.42	0.23	0	0.3
Omission errors (*n*)	−0.34	0.21	0.37	−0.06	0.09	−0.21	0.1	−0.04	0.04	−0.03
Correct answers reaction time (mean) (ms)	−0.34	0.4	0.26	−0.10	0.16	−0.31	−0.17	−0.17	−0.13	−0.24
Correct answers reaction time (SD) (ms)	−0.51 *	0.22	0.22	0.02	−0.01	−0.21	−0.19	−0.17	−0.38	−0.29
Commission errors (*n*)	−0.40	0.03	0.03	0.03	−0.16	−0.21	−0.35	−0.25	−0.19	−0.31
Normalized commission errors(*n*)	−0.47	0.09	0.04	−0.02	−0.19	−0.24	−0.33	−0.27	−0.17	−0.32

WRI = Wender–Reimherr Interview, WURS-k = Wender–Utah Rating Scale, UPPS = UPPS Impulsive Behavior Scale, GHQ-28 = General Health Questionnaire-28. *n* = number of, SD = standard deviation, ms = millisecond, cm = centimeter, s = second, m = meter, mm = millimeter, cm^2^ = square centimeter. * *p* < 0.05. ** *p* < 0.01. † *p* < 0.005. T

## Data Availability

All datasets generated for this study are included in the article.

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
