# Peer review of "A Within-Sample Comparison of Two Innovative Neuropsychological Tests for Assessing ADHD"

_brainsci, 2020, doi:10.3390/brainsci11010036_

Round 1

Reviewer 1 Report

the paper is interesting 

i wonder if the authors can improve the number of subjects because i think that the population is too small before arriving to a definite result and suggest that the 2 tests can be used, even if the results are absolutely interesting

Reviewer 2 Report

I read the paper:  a within-sample comparison of two innovative neuropsychological tests for diagnosing ADHD 
This paper reports a study about the interest of the use of two innovative neuropsychological tests to assess ADHD.
To my mind it is important to underlign that ADHD is a clinical diagnosis. it is necessary to have the developmental history of the subject. 
It is important to develop tools to assess and to evaluate, notably after treatment.
Thus, the title must be change   (not diagnosing but assess)
The originality of this study is the use of neuropsychological test using virtual reality. There are few validated tools in the area of adult ADHD. It is then important to conduct this kind of study even if few subjects have been included.

I have questions about the description of the population:
- Do we know if patients have been diagnosed ADHD in childhood
- We know that patients did not receive pharmacological treatments during the evaluation.. But what about the pharmacological status of subjects before the protocol (if they received treatment: when they have stopped treatment? And which kind of treatments?

Moreover because patients  presented comorbidities  if would be interesting to explore the results taking account these variables. 
